# High-risk HPV genotypes in Zimbabwean women with cervical cancer: Comparative analyses between HIV-negative and HIV-positive women

Oppah Kuguyo[1,2], Racheal S. Dube Mandishora[3,4], Nicholas Ekow Thomford[2,5], Rudo Makunike-Mutasa[6], Charles F. B. Nhachi[1], Alice Matimba[7], Collet Dandara[2]*

1 Department of Clinical Pharmacology, University of Zimbabwe College of Health Sciences, Harare, Zimbabwe, 2 Division of Human Genetics, Department of Pathology, Pharmacogenomics and Drug Metabolism Group, Institute of Infectious Diseases and Molecular Medicine, University of Cape Town, Observatory, Cape Town, South Africa, 3 Faculty of Health Sciences, Department of Medical Microbiology Unit, University of Zimbabwe College of Health Sciences, Harare Zimbabwe University of Zimbabwe, Medical Microbiology Unit, Harare, Zimbabwe, 4 Early Detection, Prevention and Infections Branch, International Agency for Research on Cancer, Lyon, France, 5 Department of Medical Biochemistry, School of Medical Sciences, College of Health and Allied Sciences, University of Cape Coast, Cape Coast, PMB, Ghana, 6 Department of Pathology, University of Zimbabwe College of Health Sciences, Harare, Zimbabwe, 7 Advanced Courses and Scientific Conferences, Wellcome Genome Campus, Hinxton, United Kingdom

* collet.dandara@uct.ac.za

**Data Availability Statement:** The authors recognise the importance of data sharing in

## Abstract

### Background

High-risk human papillomavirus HPV (HR-HPV) modifies cervical cancer risk in people living with HIV, yet African populations are under-represented. We aimed to compare the frequency, multiplicity and consanguinity of HR-HPVs in HIV-negative and HIV-positive Zimbabwean women.

### Methods

This was a cross-sectional study consisting of women with histologically confirmed cervical cancer attending Parirenyatwa Group of Hospitals in Harare, Zimbabwe. Information on HIV status was also collected for comparative analysis. Genomic DNA was extracted from 258 formalin fixed paraffin embedded tumour tissue samples, and analysed for 14 HR-HPV genotypes. Data was analysed using Graphpad Prism and STATA.

### Results

Forty-five percent of the cohort was HIV-positive, with a median age of 51 (IQR = 42–62) years. HR-HPV positivity was detected in 96% of biospecimens analysed. HPV16 (48%), was the most prevalent genotype, followed by HPV35 (26%), HPV18 (25%), HPV58 (11%) and HPV33 (10%), irrespective of HIV status. One third of the cohort harboured a single HPV infection, and HPV16 (41%), HPV18 (21%) and HPV35 (21%) were the most prevalent. HIV status did not influence the prevalence and rate of multiple HPV infections

validation, replication and re-analysis, to expand the current body of knowledge and optimise public health. In the current approved ethics protocol for the study, no consent from the study participants was obtained for data sharing. To maintain ethical research practice, the authors will make the data accessible with permission from local ethical review boards, namely the Joint Research Ethics Committee for the University of Zimbabwe College of Health Sciences and The Parirenyatwa Group of Hospitals (citing protocol number: JREC/412/16), as well as Medical Research Council of Zimbabwe (citing reference number: MRCZ/A/2153) on the contact details given below. Seeking ethical approval and engaging the local institutional review boards for re-analysis, validation and replication is important to safeguard the participant information and to ensure governed and harmonised use of this data. Contact details: Postal address: Joint Research Ethics Committee for the University of Zimbabwe College of Health Sciences and The Parirenyatwa Group of Hospitals, Box A178, Avondale, Harare, Zimbabwe. Tel: +2634708140. E-mail: jrec.office@gmail.com or jrec@medsch.uz.ac.zw Postal address: Medical Research Council of Zimbabwe, Cnr Josiah Tongogara/Mazowe Str, Harare, Zimbabwe. Tel: +2634791792 or +2634791193. E-mail: mrcz@mrcz.org.zw.

**Funding:** The Organisation for Women in Science in the Developing World (OSWD), Medical Research Council of South Africa (SAMRC), the National Research Foundation (NRF) of South Africa and the University of Cape Town provided support in the form of travel funding for OK to conduct work in CD's research group. No additional external funding was received for this study.

**Competing interests:** The authors have declared that no competing interests exist.

(p>0.05). We reported significant (p<0.05) consanguinity of HPV16/18 (OR = 0.3; 95% CI = 0.1–0.9), HPV16/33 (OR = 0.3; 95% CI = 0.1–1.0), HPV16/35 (OR = 3.3; 95% CI = 2.0–6.0), HPV35/51 (OR = 6.0; 95%CI = 1.8–15.0); HPV39/51 (OR = 6.4; 95% CI = 1.8–15), HPV31/52 (OR = 6.2; 95% CI = 1.8–15), HPV39/56 (OR = 11 95% CI = 8–12), HPV59/68 (OR = 8.2; 95% CI = 5.3–12.4), HPV66/68 (OR = 7; 95% CI = 2.4–13.5), independent of age and HIV status.

## Conclusion

We found that HIV does not influence the frequency, multiplicity and consanguinity of HR-HPV in cervical cancer. For the first time, we report high prevalence of HPV35 among women with confirmed cervical cancer in Zimbabwe, providing additional evidence of HPV diversity in sub-Saharan Africa. The data obtained here probes the need for larger prospective studies to further elucidate HPV diversity and possibility of selective pressure on genotypes.

## Introduction

Human papillomavirus (HPV) is a ubiquitous virus, which, under normal physiological conditions can be cleared within 24 months [1,2]. In the presence of favourable physiological and behavioural risk factors, persistent HPV infections lead to carcinogenesis in the cervix and other organs, especially in immunocompromised individuals [3–7]. Numerous HPV genotypes with phylogenetic relatedness have been discovered, yet these genotypes exhibit distinct pathogenic aptitude. High risk HPVs (HR-HPVs) such as 16, 18, 31, 33, 39, 45, 51, 52, 56, 58, 59 and 68 harbour high oncogenic potential, and so, have been influential in the development of HPV prophylactic vaccines [8,9]. The coverage for the HPV prophylactic vaccines is in its infancy in most developing countries, where the coverage is estimated to be more than 12-fold lower than developed countries [10]. As a matter of interest countries such as Zambia, Malawi and Zimbabwe, where the highest global cervical cancer cases are recorded, are still in the pilot and early implementation phases of the national HPV vaccination programs [11,12].

Evidences from diverse global HPV ethnogeography data patterns, for example, shows that HPV51 is predominantly observed in Caucasians, while HPV58 is skewed towards East Asia and Latin America [13–17]. An abundance of data also substantiates that Africa harbours significant heterogeneity between populations, and to be specific, persistent HPV66 is ubiquitous in Northern Africa, while genotypes such as HPV35 and 52 are reportedly abundant in sub-Saharan Africa (SSA) [18–23]. It is thought that the stark dissimilarities of HPV distribution and cervical cancer between SSA and the rest of the world are influenced by HIV co-infection, leading to the classification of cervical cancer as an AIDS-defining condition [5,24–29]. However, emerging data to this effect has been contraindicative. Of great relevance is the unabated cervical cancer burden even after the introduction of anti-retroviral therapies (ART), compared to other HIV-related cancers, that have since dwindled [30–39]. This data suggests other potential competing risk factors, which are yet to be defined.

Despite the strides made to understand the relationship between HPV, HIV and cervical cancer, there is still a gap in SSA. Particularly, in Zimbabwe, most studies have examined HPV in normal cervical cytology and precancerous lesions, leaving only a few studies that focus on cervical tumour tissue [5,7,17,40–43]. High prevalence of genotypes such as HPV16, 18, 31, 33,

45, 56 have been detected in tumour tissue non-synonymously [22,44–46]. There is very little data on the HPV genotype profile for cervical cancer in Zimbabwe. This study aimed to analyse the frequency and multiplicity of HR-HPV genotypes in HIV-negative and HIV-positive Zimbabwean women with histologically confirmed cervical cancer.

## Materials and methods

### Study design and setting

This was a cross-sectional study consisting of 258 women with histologically confirmed invasive cervical cancer diagnosis. This study recruited participants between July 2016 and January 2019 from an outpatient oncology facility, Parirenyatwa Group of Hospitals Radiotherapy and Chemotherapy Centre (RTC), in Harare, Zimbabwe. The biospecimens for analysis were collected at the pathology laboratory of diagnosis, namely Lancet Pathology and Parirenyatwa Group of Hospitals Pathology Department. Collected samples were processed as Formalin fixed paraffin embedded (FFPE) blocks.

### Ethical approval

All human health research was in accordance with the Helsinki declaration. Ethical approval was obtained from University of Zimbabwe College of Health Sciences and Parirenyatwa Group of Hospitals Joint Research Ethics Committee (412/16), Medical Research Council of Zimbabwe (A/2153), University of Cape Town Research Ethics Committee and Research Council of Zimbabwe (No: 03351).

### Study participants

The participants from the current study were nested in a pharmacogenomics of cervical cancer study, aiming to recruit women who were newly diagnosed with cervical cancer, and were eligible to receive radical therapy. Potential study participants were identified through the RTC hospital registry. To be considered eligible for this study, women had to be ≥18 years, with histologically confirmed diagnoses of invasive cervical cancer staged between 1b – 4A, and were earmarked for curative anti-cancer therapy. Individuals who were diagnosed with resectable (stage 1A) disease that did not require chemo/radio- therapies or advanced disease that required palliative care (stage 4B) were excluded from the study, because these individuals were referred for care outside of the RTC. Eligible participants were informed of the study verbally, and the women willing to enroll provided written consent. To maintain participant confidentiality, all participant data was de-identified and assigned study numbers. The clinical and demographic information for all recruited participants were abstracted onto a case report form. Demographic information such as age, residency, histopathological tumour characteristics, were collected from the patient folder in the RTC. In addition, behavioral, lifestyle factors and sexual history were collected by interviewing the study participants, namely history of alcohol, smoking, parity, age at sexual debut, number of sexual partners, history of circumcised partner, sexually transmitted infections were also collected. Data on HIV status was also collected for comparative analysis.

### DNA extraction and HPV genotyping

A total of 5mg of the FFPE blocks were sectioned off in preparation for genetic analysis. Genomic DNA was isolated from the FFPE scrolls following a slightly modified manufacturers' protocol from Zymo genomic DNA extraction FFPE kit (Zymo Research, California, USA). Paraffin was removed by adding a Xylene-based deparaffinization solution and incubating at

55°C for 30 minutes. The wax layer was then aspirated and the remaining biopsy tissue was digested overnight using 10mg/ml Proteinase K (Zymo Research, California, USA). The quantity and quality of DNA was assessed using a Nanodrop™ Spectrophotometer 2000/2000c (Nano-drop™, ThermoFisher, Denver, USA) and DNA integrity was evaluated by running a 1% agarose gel electrophoresis. The extracted DNA was stored at -80°C for further analysis.

Genotyping for high-risk HPV subtypes was undertaken using clinically validated Anyplex™ II HPV HR detection kit (Seegene, Seoul, South Korea). The Anyplex™ II HPV HR detection kit (Seegene, Seoul, South Korea) targets the HPV L1 gene (encoding the capsid) using primers provided in the Anyplex kit. This assay uses Dual Priming Oligonucleotides (DPO™) and tagging oligonucleotide cleavage and extension (TOCE™) technology which enables the distinguishing of 14 HR-HPVs in a single reaction, namely, HPV 16, 18, 31, 33, 35, 39, 45, 51, 52, 56, 58, 59, 66, 68. A total reaction of 20µl was used during genotyping, following the manufacturers' protocol. Each reaction contained a human β-globin internal control, L1 primers, oligonucleotides and enzymes. A negative control, as well as 3 positive controls (DNA mixture of pathogen clones) were run concurrently with each test plate to confirm the validity of the amplification. The CFX96™ real-time thermocycler (Bio-Rad, Applied Biosystems, California, USA) was used under thermocycling conditions recommended in the Seegene protocol. Fluorescence was detected using the end point cyclic melting temperature analysis. Seegene Viewer v2.0 program (Seegene, Seoul, South Korea) was used to analyse and interpret fluorescence data. Each result was considered valid when the internal control was detected in the sample (IC+). Positive result (+) indicated HPV DNA presence, while negative result (-) indicated the absence of the viral DNA. All HPV negative results with a positive endogen internal control (IC+) were not re-analysed.

## HPV genotype frequency data mining

Data on the HPV genotypes of African populations was extracted from the HPV Information Centre database [47]. The HPV Information Centre consolidates data from published literature and official reports published by the World Health Organisation, United Nations, The World Bank, International Agency for Research on Cancer's Globocan and Cancer Incidence in Five Continents. We filtered the report for data on the African continent, and collected the data on the most frequently detected HPV genotypes in women with confirmed cervical cancer.

## Statistical and data analysis

STATA v12.0 (StataCorp LLC, Station College, TX, USA) and Graphpad v8 (Prisma, San Diego, California) statistical software packages were used for data analysis. Demographic and clinical data were allocated into either continuous data or categorical data. Continuous data were expressed as mean ± standard deviation or median (inter-quartile range), while the categorical data was expressed as absolute or relative frequencies. For the comparison of the socio-demographic factors between HIV negative and HIV positive study participants, the two-sample Wilcoxon rank-sum test and Chi-squared test of independence was used. Correlation between the various HPV genotypes and HIV status was performed using Poisson's and Logistic regression. The Chi-squared test of homogeneity was used to compare the frequency of HPV genotypes among African women using Fisher's exact and Chi-squared values as test statistics. Univariate regression was used to determine an association between the established HPV risk factors with the HR-HPV genotype infections. Multivariate logistic regression was used to determine an association between the pairings of the HPV genotypes using age, and HIV status as predictors for the model. Further sensitivity analyses were performed for the co-

segregating HPV genotypes, using multivariate regression models, and controlling for known HPV risk factors, namely, age, history of STI, parity, HIV status, age of sexual debut and tumour histology. We considered p<0.05 as statistically significant. For analysis, all individual HPV genotypes regardless of whether they occur as single or multiple infections were analysed and reported as standalone genotypes. None of the statistical analysis took into account the species from which the HPV genotypes are from.

## Results

### Sociodemographic features of the study participants

Sociodemographic and clinical characteristics of the study group are summarized in Table 1. Of the 258 cervical cancer patients recruited for this study, 45% (n = 116) were confirmed HIV positive. The median age (inter-quartile range) of the participants was 51 years (42–62). The overall median for age was significantly different (p = 0.001) between the HIV-positive and HIV-negative cervical cancer groups. Further analysis of the age frequency distribution showed that in HIV positive women, age was skewed to the younger women (<50 years) compared to women with HIV-negative status (Fig 1). Most of the sociodemographic risk factors were comparable between the groups, however, the behavioural risk factors, such as number of sexual partners (p = 0.001) and history of sexually transmitted infections (p = 0.010) differed between the HIV-negative and HIV-positive groups. In univariate regression analysis, none of the HPV-related risk factors (i.e., age, sexual debut, parity and STI history) were found to be significantly associated with any of the HR-HPV genotypes or multiple HR-HPV genotype infections (S1 Table).

### HPV genotype prevalence

The 258 cervical cancer patients were screened for HR-HPV DNA, and 96% (n = 248) were positive, representing 14 HPV types (16, 18, 31, 33, 35, 39, 45, 51, 52, 56, 58, 59, 66 and 68) (Table 2) There was no difference in HPV positivity and HPV type between the HIV negative and positive groups. Distribution of HPV in this cohort stratified by HIV status is illustrated in Table 2. HPV16 was the most prevalent genotype detected in 48% (n = 123) of the participants, with high frequencies for the following genotypes; HPV35 (26.4%), HPV18 (25.2%), HPV58 (10.5%), HPV33 (9.7%), and HPV31 (6.6%).

### HPV mono-infections

About one-third (n = 85) of the study cohort were harbouring a single HR-HPV infection. The prevalence and distribution of HR-HPV mono infections is illustrated in Fig 2. In this group of women with confirmed cervical cancer, the most commonly reported mono-infections were HPV16 (41%), HPV18 (21%) and HPV35 (21%). This study did not observe any HPV mono-infections arising from HPV genotypes 56, 59 and 66. Comparative analysis between HIV-negative and HIV-positive women showed that there were no statistically significant differences in the prevalence HR-HPV mono-infections (p = 0.400).

### Multiple HR-HPV infections

Majority (67%) of the cohort exhibited multiple HPV infections. Of the 14 HR-HPV genotypes analysed in this study, 63% (n = 163) harboured multiple (>2) HPV genotypes (Table 3). Fifty nine percent of the study participants exhibited infection with 2 HR-HPVs, while 19% were infected with 3 HR-HPVs. The most common multiple infections included HPV16 and 35 (21%), followed by HPV16/18 and HPV16/58. The highest number of multiple HPV

**Table 1. Sociodemographic and behavioural characteristics of the study participants.**

| | Combined | HIV- negative | HIV-positive | p-value |
|---|---|---|---|---|
| **Total number of participants, N(freq)** | 258 (1.00) | 142 (0.55) | 116 (0.45) | |
| **Median Age (IQR)** | 51 (42–62) | 56 (45–64) | 46 (41–54) | **0.001**[a] |
| **Median Parity (IQR)** | 4 (3–6) | 5 (4–7) | 2 (3–5) | **0.001** [a] |
| **Median Sexual partner history (IQR)** | 1 (1–3) | 1 (1–1) | 2 (1–3) | **0.001** [a] |
| **Median Age at sexual debut (IQR)** | 17 (15–19) | 17 (15–19) | 17 (15–20) | 0.525 [a] |
| **Residency** | | | | |
| Urban | 137 (0.53) | 77 (0.54) | 60 (0.52) | **Ref.** |
| Peri-urban | 48 (0.19) | 27 (0.19) | 21 (0.18) | 0.438 [b] |
| Rural | 73 (0.280) | 38 (0.27) | 35 (0.30) | 0.392 [b] |
| **Marital Status** | | | | |
| Married | 114 (0.44) | 76 (0.54) | 68 (0.59) | **Ref.** |
| Never married | 9 (0.03) | 3 (0.02) | 6 (0.05) | 0.418 [b] |
| Divorced/ | 51 (0.20) | 25 (0.18) | 22 (0.19) | 0.961 [b] |
| Widow | 84 (0.33) | 38 (0.27) | 22 (0.19) | 0.167 [b] |
| **Alcohol consumption** | | | | |
| No | 229 (0.89) | 122 (0.86) | 107 (0.92) | **Ref.** |
| Yes | 29 (0.11) | 16 (0.14) | 6 (0.08) | 0.080 [b] |
| **Smoking History** | | | | |
| No | 246 (0.97) | 132 (0.93) | 114 (0.98) | **Ref.** |
| Yes | 12 (0.03) | 6 (0.07) | 6 (0.02) | 0.804 [b] |
| **STI history** | | | | |
| No | 181 (0.70) | 109 (0.77) | 72 (0.62) | **Ref.** |
| Yes | 77 (0.298) | 33 (0.23) | 44 (0.38) | **0.010** [b] |
| **Circumcised partner** | | | | |
| No | 232 (0.90) | 123 (0.87) | 108 (0.93) | **Ref.** |
| Yes | (0.10) | 19 (0.13) | 8 (0.07) | 0.091[b] |
| **Tumour Histology** | | | | |
| Squamous Cell | 207 (0.80) | 115 (0.83) | 92 (0.77) | **Ref.** |
| Adenocarcinoma | 25 (0.10) | 14 (0.10) | 11 (0.09) | 0.950[b] |
| Adenosquamous | 11 (0.04) | 2 (0.01) | 9 (0.08) | **0.017** [b] |
| Other* | 15 (0.06) | 8 (0.06) | 7 (0.06) | 0.900[b] |

**IQR** = inter-quartile range;

[a] = Wilcoxon rank sum test**;**

[b] = Chi-squared test;

Other* = spindle cell carcinoma, papillary serous carcinoma, adenoid cystic, adenosarcoma, small cell carcinoma.

serovariants per individual reported was 6, in 3.7% of study subjects (Table 3). There was no difference in the number of multiple infections between HIV positive and HIV negative cervical cancer patients. Distribution of multiple HPV genotypes stratified by HIV status are illustrated in S2 Table.

Furthermore, the incidence of multiple-type HPV infections was assessed to determine if specific HPV genotypes tend to co-segregate, or if the multiple-type infections occur as a result of chance. HIV status was not a significant contributor to the co-segregation patterns. We used multivariate logistic regression, using the HPV genotypes as predictors, while controlling for age and HIV status, as illustrated below (Fig 3). We detected statistically significant associations (p<0.05) between HPV16 and 33 (OR = 0.3; 95% CI = 0.1–1.0), HPV16 and 35

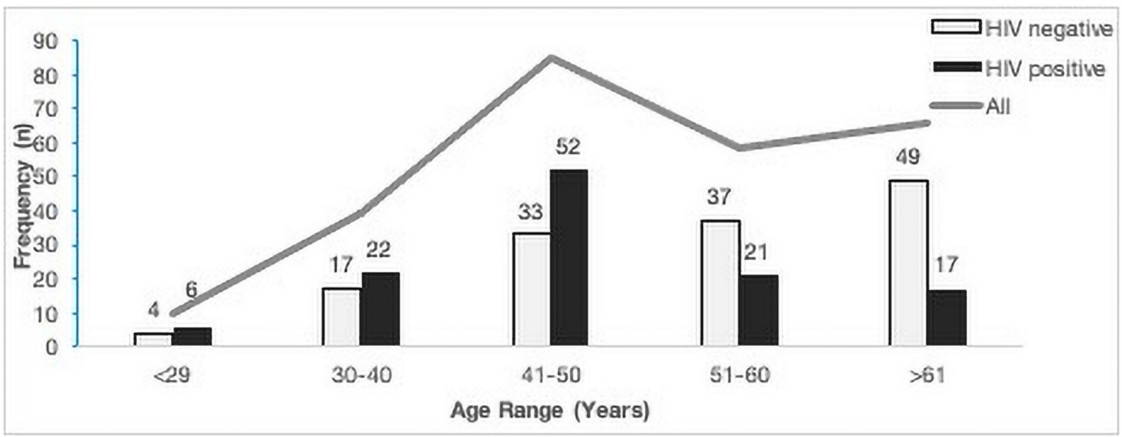

**Fig 1. Age distribution of study cohort stratified by HIV status (n = 258).**

(OR = 3.3; 95% CI = 2.0–6.0), HPV16 and 18 (OR = 0.3 95% CI = 0.1–0.9), HPV35 and 51 (OR = 6.0; 95%CI = 1.8–15.0), HPV39 and 51 (OR = 6.4; 95% CI = 1.8–15), HPV31 and 52 (OR = 6.2; 95% CI = 1.8–15), HPV39 and 56 (OR = 11 95% CI = 8–12), HPV59 and 68 (OR = 8.2; 95% CI = 5.3–12.4), and HPV66 with 68 (OR = 7; 95% CI = 2.4–13.5). The respective odds ratios and 95% confidence intervals for all significant HPV genotypes are shown in Fig 4. In order to ensure that lack of significance observed was not a result of low study power, a sensitivity appraisal was conducted, using multivariate regression analyses controlling for age, history of sexually transmitted infections, parity, HIV status, age at sexual debut and

**Table 2. HR-HPV genotypes by HIV status (n = 258) and the HPV phylogenetic classification.**

| HPV genotype | Frequency (Proportion) | | | IRR (95% CI) | p value |
|---|---|---|---|---|---|
| | All | HIV negative | HIV positive | | |
| Negative | 10 (0.04) | 6 (0.02) | 4 (0.02) | 0.40 (0.22–1.00) | 0.443 |
| 16[a] | 123 (0.48) | 66 (0.26) | 57 (0.22) | 0.97 (0.69–1.37) | 0.881 |
| 18[b] | 65 (0.25) | 34 (0.13) | 31 (0.12) | 1.11 (0.78–1.57) | 0.573 |
| 31[a] | 17 (0.07) | 8 (0.03) | 9 (0.04) | 1.06 (0.57–1.96) | 0.854 |
| 33[a] | 25 (0.10) | 18 (0.07) | 7 (0.03) | 0.81 (0.49–1.34) | 0.416 |
| 35[a] | 68 (0.26) | 40 (0.16) | 28 (0.10) | 0.90 (0.63–1.29) | 0.574 |
| 39[b] | 10 (0.04) | 3 (0.01) | 7 (0.03) | 1.42 (0.67–3.05) | 0.361 |
| 45[b] | 14 (0.05) | 8 (0.03) | 6 (0.02) | 1.05 (0.53–2.06) | 0.892 |
| 51[c] | 11 (0.04) | 5 (0.02) | 6 (0.02) | 1.35 (0.75–2.45) | 0.316 |
| 52[a] | 9 (0.04) | 5 (0.02) | 4 (0.02) | 0.88 (0.33–2.38) | 0.799 |
| 56[d] | 6 (0.02) | 2 (0.0) | 4 (0.02) | 1.11 (0.41–2.99) | 0.843 |
| 58[a] | 23 (0.11) | 9 (0.05) | 14 (0.05) | 1.28 (0.78–2.09) | 0.334 |
| 59[b] | 3 (0.01) | 0 (0.00) | 3 (0.01) | 1.67 (0.53–5.24) | 0.380 |
| 66[d] | 1 (0.0) | 0 (0.00) | 1 (0.01) | 1.1 (0.15–7.89) | 0.922 |
| 68[b] | 7 (0.03) | 2 (0.01) | 5 (0.02) | 1.23 (0.51–3.01) | 0.645 |

IRR = incidence rate ratio computed from the Poisson regression analysis.

[a] = Human papillomavirus alpha 9 species;

[b] = Human papillomavirus alpha 7 species;

[c] = Human papillomavirus alpha 5 species;

[d] = Human papillomavirus alpha 6 species.

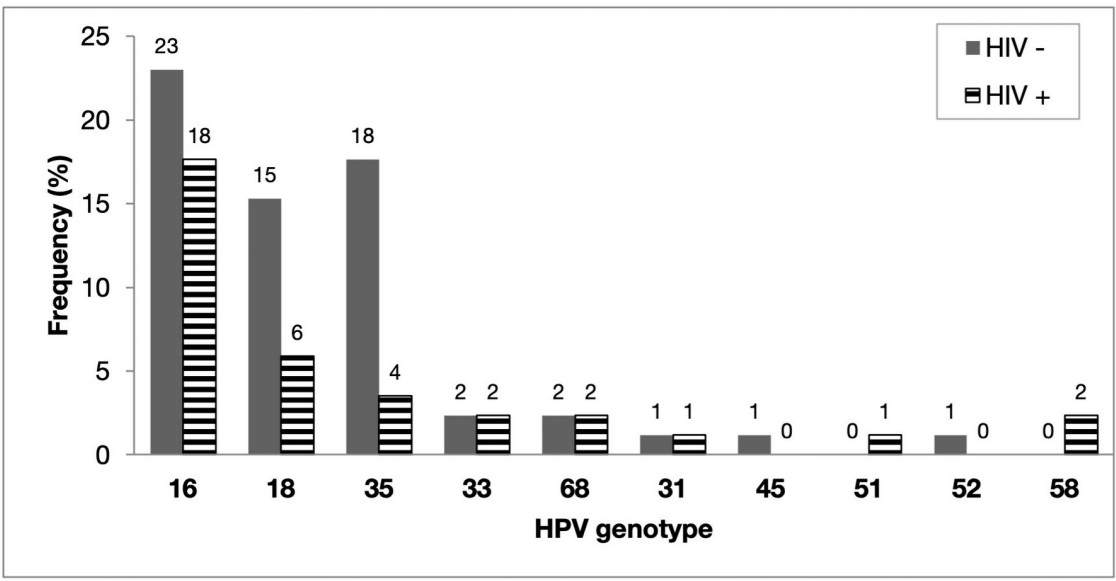

**Fig 2. The HPV mono infection distribution in women with confirmed cervical cancer (n = 85).**

tumour histology. In this sensitivity analyses, all the co-segregation patterns remained statistically significant except HPV16/33 (OR = 0.4; 95% CI = 0.2–1.9; p = 0.05), HPV35/52 (OR = 0.5; 95% CI = 0.1–4.3; p = 0.50) (S3 Table).

## Comparing the distribution of different HPV types in women from populations across Africa

The prevalence of the various HPV genotypes in the Zimbabwean study cohort was compared to the most common HPV genotypes on other African women across the continent, using data obtained from HPV Information Centre database [47] and summarized into Table 4. In comparison to the rest of the African continent data, our study reported significantly higher HPV35 (p<0.001), HPV51 (p = 0.043), HPV58 (p = 0.010) and HPV39 (p = 0.043). Across the different African regions, most of the HPV genotypes were comparable (p>0.05), and HPV16 was the most prevalent genotype. However, in North Africa (p = 0.047), the frequency of HPV16 is significantly higher, while HPV45 (p = 0.011) and HPV59 (p = 0.005) are significantly higher in West Africa compared to the other African regions. Additionally, the HPV information centre database also showed that HPV26 was one of the top 10 genotypes detected in cervical cancer tissue in Southerrn Africa, and HPV53 in Northern and Southern Africa.

**Table 3. Multiple HR-HPV infections, stratified by HIV status (n = 163).**

| Number of HPVs found together | Frequency (Proportion) | | | | |
|---|---|---|---|---|---|
| | All | HIV negative | HIV positive | OR (95% CI) | p-value |
| 2 | 94 (0.59) | 45 (0.28) | 50 (0.31) | 1.65 (0.46–5.98) | 0.443 |
| 3 | 31 (0.19) | 16 (0.10) | 15 (0.09) | 2.10 (0.55–8.01) | 0.277 |
| 4 | 19 (0.12) | 11 (0.07) | 8 (0.05) | 0.80 (0.16–4.02) | 0.782 |
| 5 | 12 (0.08) | 6 (0.04) | 6 (0.04) | 1.75 (0.08–36.28) | 0.718 |
| 6 | 6 (0.04) | 4 (0.03) | 2 (0.01) | 1.2 (0.34–4.41) | 0.756 |

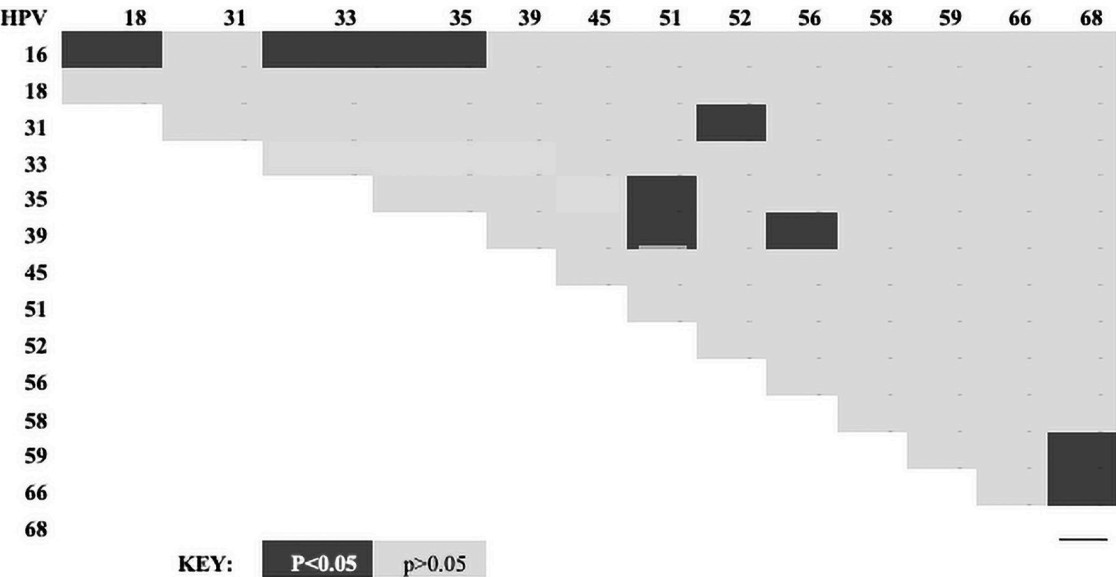

**Fig 3. Heatmap of co-segregation of any two HPV types among Zimbabweans.**

## Discussion

There is a plethora of studies that have analyzed HPV genotyping profiles in different populations across the globe, and what is apparent is that HPV16 and 18 are highly prevalent in cervical cancer. The global HPV prevalence reports HPV16 (55%) as the highest followed by HPV18 (14%), which is supported by our data, with 48% and 25%, respectively. However, in stark contrast to the global data, we report here a >13-fold higher frequency (26%) of HPV35 among Zimbabwean women with cervical cancer, compared to Caucasians (~2%) and Asian (~1%) cervical cancer patients [48,49]. Across the African continent, there is heterogeneity in the frequency of HPV35, ranging from as low as 6% in South Africa, Burkina Faso (9%), Nigeria (9%), Tanzania (15%), Mozambique (17%), and up to the comparable 24% observed among Malawian women [18,28,50–58]. It is important to note, an earlier study among Zimbabwean women [22] which showed a frequency of 11% for HPV35. This discrepancy could

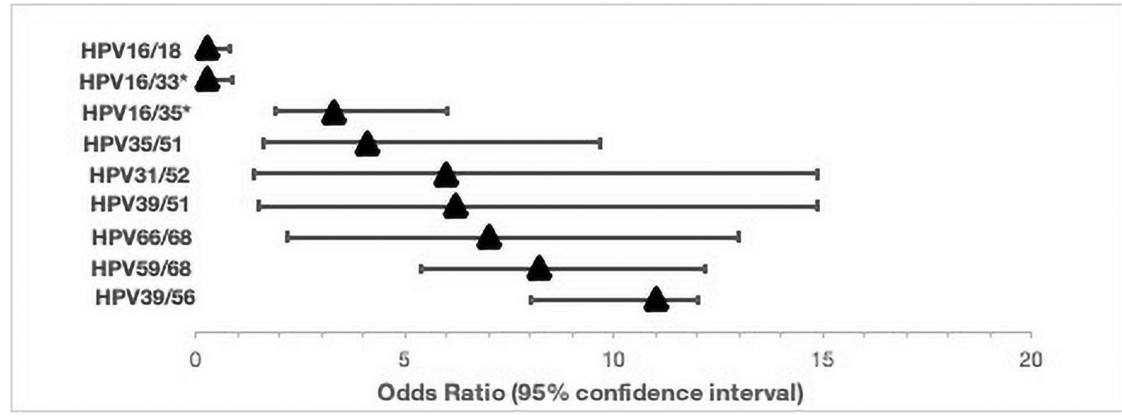

**Fig 4. Odds ratios and 95% confidence intervals (95% CI) for co-occurrence of high-risk HPV infections among Zimbabweans.**
* Indicates co-segregation of HPV genotypes in the same phylogenetic clades.

Table 4. Comparison of the frequencies of the most common HPV genotypes reported across Africa.

| | Frequency in % | | | | | | | | |
|---|---|---|---|---|---|---|---|---|---|
| | Our Study* | East Africa | West Africa | North Africa | Southern Africa | P¹ | P² | P³ | P⁴ |
| HPV16 | 48 | 50 | 36 | 62 | 48 | 0.777 | 0.086 | **0.047** | 0.999 |
| HPV18 | 25 | 18 | 20 | 17 | 15 | 0.228 | 0.397 | 0.165 | 0.077 |
| HPV31 | 7 | 2 | 9 | 3 | 3 | 0.088 | 0.088 | 0.602 | 0.194 |
| HPV33 | 10 | 6 | 3 | 3 | 7 | 0.279 | 0.297 | 0.053 | 0.047 |
| HPV35 | 26 | 4 | 4 | 3 | 6 | **<0.001** | **<0.001** | **<0.001** | **<0.001** |
| HPV45 | 5 | 9 | 16 | 9 | 6 | 0.268 | **0.011** | 0.268 | 0.756 |
| HPV51 | 4 | 3 | 0 | 0 | 0 | 0.700 | **0.043** | **0.043** | **0.043** |
| HPV52 | 4 | 4 | 2 | 3 | 2 | 0.999 | 0.407 | 0.700 | 0.407 |
| HPV56 | 2 | 0 | 2 | 0 | 0 | 0.155 | 0.999 | 0.155 | 0.155 |
| HPV58 | 11 | 3 | 2 | 3 | 1 | **0.027** | **0.010** | **0.027** | **0.029** |
| HPV59 | 1 | 0 | 10 | 0 | 0 | 0.316 | **0.005** | 0.316 | 0.316 |
| HPV66 | 0 | 0 | 0 | 2 | 0 | - | - | 0.155 | - |
| HPV39 | 4 | 1 | 0 | 0 | 0 | 0.174 | **0.043** | **0.043** | **0.043** |
| HPV53 | NA | 0 | 0 | 2 | 1 | - | - | - | - |
| HPV26 | NA | 0 | 0 | 0 | 2 | - | - | - | - |

*Reference group for comparative analaysis;

- = p-value cannot be computed; NA = not assayed in our study population; East Africa v our study data;

² = West Africa v our study data;

³ = North Africa v our study data;

⁴ = Southern Africa v our study data.

be parenthetic to methodological differences in HPV genotyping. Specifically, our study utilised the multiplex PCR method, while Mudini *et al.* used PGMY09/11 PCR and dot plot hybridization. Although both methods yield highly concordant findings, the PGMY09/11 HPV detection method has been associated with a higher rate of false negatives compared multiplex PCR [59–61].

Moreover, a high HPV genotype frequency in the general population is not an adequate proxy for carcinogenic potential. For example, in Latin America, HPV35 is as ubiquitous as in SSA, but is widespread in precancerous lesions and not in cancer [62]. Novel genetic variations in the coding and non-coding regions of HPV35 have been discovered, especially in African women, and women of African ancestry, which are not present in Caucasian, Asian and Hispanic populations [23,63]. These genetic polymorphisms are thought to account for the distinct HPV35 virulence and oncogenic potential observed in Africans, versus Latin America. However, not much whole HPV35 sequencing has been reported on African cervical tumour samples, thus, there is a potential to discover even more variants in different African populations.

While it has been previously thought that HPV16 and a few other genotypes exhibited higher innate potential to evade the host immune system, and other genotypes required a suppressed immune system to flourish *e.g.* HPV35 [64,65], new data seem to show that infection with HPV35 can also be correlated with persistence and unresolvable precancerous lesions [13,23,28,53,66–68]. For example, and contrary to earlier expectations, studies from SSA show that HPV35 is one of the most ubiquitious genotypes, irrespective of HIV status, further confirming our observation in the present study [19,20,22,57,69]. Our observations agree with reports from other studies evaluating HPV prevalence, distribution and multiplicity in

Zimbabwean women, notwithstanding cervical cytology status [22,63]. It is equally vital to include in the equation immune reconstitution resulting from increased access to ART in Zimbabwe, which could be contributing to the homogeneity of the HPV genotype distribution and observed multiplicity in our study, and other Zimbabwean studies.

Surprisingly, the introduction of ART has not been met with sizeable decline of cervical cancer, compared to other HIV-related malignancies such as Kaposi's Sarcoma and Non-Hodgkins' Lymphoma, which are now rarely detected [70,71]. To ascertain the role of immune reconstitution in HPV persistence and cervical cancer, immunological factors (which were not evaluated in the current study) such as HIV viral load, HPV viral load, CD4+ count and duration of ART need to be investigated [53,72]. There is, however, evidence in other African populations to suggest that only a small fraction of women with HIV/HPV are likely to develop invasive cervical cancer without influence from CD4+ count, thus pointing to an increased role of the other competing risk factors and the need to interrogate them as potential HPV persistence drivers, including host genetics [73–82].

Persistence of HPV and the presence of multiple HPV infections are both phenomena closely correlated with HIV-induced immunosuppression [83–88]. Additionally, other sexual behavioural characteristics such as history of STIs, higher number of sexual partners are also key contributors for HPV multiplicity [89–91]. Although our study reports high multiple HPV infection rate, we found no relationship between sexual behavioral factors such as HIV and STI history, and risk of harbouring multiple HPV genotypes. Not many studies have sought to understand the patterns by which multiple HPV genotypes co-segregate and interact to induce carcinogenesis. Some studies highlight that multiple infections occur as random events, while other studies allude to competitive or co-operative consanguinity of specific HPV genotypes [59,92–94]. Aggregation of HPV genotypes with phylogenetic relatedness has been observed at different degrees, and HPV 16, 18, 58 and 66 have been shown to occur mostly as single infections, in Americans and Latin Americans, while HPV35 and 45 were more likely to present as multiple infections [59,92–94]. On the contrary, our study reports no statistically significant differences in the HPV genotype distribution for single and multiple infections for any of the HPV genotypes. Furthermore, we report type-specific HPV clustering of genotypes from the same phylogenetic clades, namely, HPV16/33 and HPV16/35; while the rest of the co-segregation we reported here was of genotypes that are of diverse phylogenetic clades. Previous data suggests that multiple infection with genotypes of diverse HPV phylogenetic clades is directly correlated with no history of the HPV prophylactic vaccine, ascribed to the punitive host induced cross-protection, compared to the vaccine induced response [59,92,95–98]. In Zimbabwe, the HPV vaccination program is still in early implementation phase, therefore, most women at reproductive ages and above are reliant on host immunology prevention. This is similar in most of Africa, where there is limited access and coverage of the HPV vaccine, even though the burden of cervical cancer is disproportionately high here, and the populations are exposed to other dissonant evolutionary selective pressures which collectively exacerbate the risk of HPV persistence [99,100].

Analysis of the HPV genotypes detected across Africa, by region confirms distinct HPV ethnogeographical patterns. Of great interest is the probable high-risk genotype, HPV26, which was observed only in cervical cancer tissue from Southern Africa. Because HPV26 is not a definitive HR-HPV, it is often overlooked, so its burden in Africa may be under-reported, and consequently its contribution to cervical cancer. HPV51, HPV56 and HPV59 were seemingly exclusively observed in East and West African women. It is important to note that these genotypes are not covered by current HPV vaccines. As of 01 August, 2021, the most robust vaccine, Gardasil-9, targets HPV 6, 11, 16, 18, 31, 33, 45, 52 and 58, for which the estimated cross-protective potential translates to cloistering <70% of HPV-related cancers [8,101–103]. Even

so, longitudinal data illustrates diminished cross-protective immunogenicity and reactogeni-city over time, leaving some ethnogeographical groups exposed and susceptible to HPV infec-tion [23,53,54,104–107]. Given the significant HPV diversity, and the genetic variation in Africa it may be of great benefit to conduct large-scale, longitudinal HPV genotyping studies in women from different African countries, such as the African Collaborative Center for Microbiome and Genomics Research (ACCME) study in Nigeria [107] so as to understand the impact of HPV, HIV and related host genetic factors for potential utility as biomarkers for cer-vical cancer in Africa. This large-scale empirical data would be key to determine which geno-types characteristically resolve, or progress to cervical cancer along with HIV in African settings and are fundamental towards developing a comprehensive and Africa-specific HPV vaccine.

Additionally, no HR-HPV infection detected in about 4% of the cohort irrespective of HIV status. This could have resulted from the fact that this particular sub-group were harbouring an HPV subtype that was not characterised by the assay kit used in this study. For example, HPV6 and 11 are established to play a causative role in genital warts, or HPV26 which we found in the comparative analysis to be idiosyncratic to Southern Africa [17,108]. Future stud-ies should characterise for HPV using broader spectrum kits, simultaneously detecting low- and high- risk HPV subtypes. It is also possible that this group harbours HPV-negative cervical tumours, which are known to be quite rare and occurring in <5% of cases. A previous study conducted by Kjetland and collaborators (2010) reported schistosomiasis-induced squamous intraepithelial neoplasia with no HPV on Zimbabwean women [109].

In conclusion, our study analysed HR-HPV sub-types in Zimbabwean women, and pres-ents data which will add to HPV diversity in SSA. This study is the first to describe high fre-quency of HPV35, comparable to other HR-HPVs such as HPV18 in Zimbabwean women with cervical cancer. Furthermore, this study is also the first to report multiple type-specific HPV subtypes in African populations, providing empirical evidence of high consanguinity of HR-HPV genotypes despite HIV status. The data obtained here is fundamental towards deter-mining the efficacy of the commercially available prophylactic vaccines in SSA populations that harbour disparate viral genome profiles and are subject to evolutionary pressures.

## Supporting information

**S1 Table. Risk factors of HPV in association with the various HPV genotypes.**
(PDF)

**S2 Table. The number of individuals harbouring specific HPV genotypes stratified by HIV status.**
(PDF)

**S3 Table. Sensitivity analysis for HPV co-segregation.**
(PDF)

## Acknowledgments

The authors wish to acknowledge Gamuchirai Rinashe and Frances Desales Misi who played an instrumental role in data collection and specimen retrieval.

## Author Contributions

**Conceptualization:** Oppah Kuguyo, Charles F. B. Nhachi, Alice Matimba, Collet Dandara.

**Data curation:** Oppah Kuguyo, Racheal S. Dube Mandishora, Rudo Makunike-Mutasa, Alice Matimba, Collet Dandara.

**Formal analysis:** Oppah Kuguyo, Racheal S. Dube Mandishora, Nicholas Ekow Thomford, Charles F. B. Nhachi, Alice Matimba, Collet Dandara.

**Funding acquisition:** Alice Matimba, Collet Dandara.

**Investigation:** Oppah Kuguyo, Collet Dandara.

**Methodology:** Oppah Kuguyo, Racheal S. Dube Mandishora, Nicholas Ekow Thomford, Rudo Makunike-Mutasa, Alice Matimba, Collet Dandara.

**Project administration:** Oppah Kuguyo, Alice Matimba, Collet Dandara.

**Resources:** Charles F. B. Nhachi, Alice Matimba, Collet Dandara.

**Software:** Nicholas Ekow Thomford, Collet Dandara.

**Supervision:** Nicholas Ekow Thomford, Rudo Makunike-Mutasa, Charles F. B. Nhachi, Alice Matimba, Collet Dandara.

**Validation:** Nicholas Ekow Thomford, Collet Dandara.

**Visualization:** Racheal S. Dube Mandishora, Collet Dandara.

**Writing – original draft:** Oppah Kuguyo.

**Writing – review & editing:** Oppah Kuguyo, Racheal S. Dube Mandishora, Nicholas Ekow Thomford, Rudo Makunike-Mutasa, Charles F. B. Nhachi, Alice Matimba, Collet Dandara.

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
