## [Decision Letter · Decision Letter 0]

19 Mar 2021

PONE-D-21-04071

High-Risk HPV genotypes in Zimbabwean women with cervical cancer: Comparative Analyses between HIV-negative and HIV-positive women.

PLOS ONE

Dear Dr. Dandara,

Thank you for submitting your manuscript to PLOS ONE. After careful consideration, we feel that it has merit but does not fully meet PLOS ONE’s publication criteria as it currently stands. Therefore, we invite you to submit a revised version of the manuscript that addresses the points raised during the review process.

We look forward to receiving your revised manuscript.

Kind regards,

Joakim Dillner, M.D.

Academic Editor

PLOS ONE

Journal Requirements:

'OK received PhD fellowship support from the OWSD and CD’s research group in funded by the Medical Research Council of South Africa (SAMRC), the National Research Foundation (NRF) of South Africa and the University of Cape Town'

'No'

Reviewers' comments:

Reviewer's Responses to Questions

**Comments to the Author**

1. Is the manuscript technically sound, and do the data support the conclusions?

Reviewer #1: Yes

Reviewer #2: Yes

2. Has the statistical analysis been performed appropriately and rigorously? 

Reviewer #1: No

Reviewer #2: Yes

3. Have the authors made all data underlying the findings in their manuscript fully available?

Reviewer #1: No

Reviewer #2: No

4. Is the manuscript presented in an intelligible fashion and written in standard English?

Reviewer #1: Yes

Reviewer #2: Yes

5. Review Comments to the Author

Reviewer #1: This cross-sectional study on HPV types in 258 cases of cervical cancer in Zimbabwe, categorized by HIV status, supports earlier results, which have shown that differences in HPV type distribution diminishes with the severity of the cervical lesions. This study also brings more data to the growing body of knowledge regarding the increased proportion of HPV35 in women from sub-Saharan Africa/ancestry.

1. Throughout the text “despite” is often used when it should be “irrespective of” or “regardless of” for example third line in result-section of abstract.

2. Methods: I would suggest using regression analyses for Table 2, 3 and 4 rather than Chi2. Either Poisson regression for prevalence ratios or Logistic regression for odds ratios.

3. Results: I suggest categorizing by HIV status in Figure 2.

4. Results: For better understanding of Figure 3A and Figure 3B it might be better if the absolute numbers of certain genotype combinations were presented. I lack an explanation as to how to assess the odds ratios (eg OR increased at 3.3 for the most common combination 16/35 but decreased at OR 0.3 for the second most common combination 16/18. Is 16 reference?).

5. Results: It says 58% having 2 HR-HPVs in the text but 59% in Table 3.

6. Supplementary Table 1. Suggest adding HIV as risk factor

Reviewer #2: In this hospital-based cross-sectional study, the authors examine the distribution of high-risk HPV in HIV+ and HIV- women in Zimbabwe. The study reported high prevalent of HPV 35 among women confirmed with cervical cancer, and HIV status was not associated with frequency, multiplicity and consanguinity of HR-HPV in cervical cancer. The study is well-designed and performed, and also provides important information on high-risk HPV distribution in African women.

Several aspects could be further clarified:

1. It would be informative to provide a brief description of the HPV vaccination program and uptake in Zimbabwe early in the introduction.

2. Could the authors clarify the reasons of only including cervical cancer diagnosed at stage 1b-4.

3. It seems the histopathological characteristics are available for all cases included in this study. Such information should be included in the descriptive analysis.

4. Maybe only relevant covariables to the current study are needed to be described. Some of the information collected through the project but not used for the study might not be necessary to mention, such as weight, height, comorbidities and treatment etc.

5. I figure no significant association was detected in the univariate regression between HPV-related risk factors (i.e., age, sexual debut, parity and STI history) and any HR-HPV genotypes might be due to limited power, when the HR-HPV types were stratified by each type. However, those are known factors that associated with HPV infection. Could the authors provide estimates (figure 3b) additionally adjusting for those factors and histology at least as sensitivity analysis when examine the co-occurrence of HPV types. Besides, please clarify what is the reference group for the ORs.

6. A legend explaining how the HR-HPV types were classified (ie. hierarchical classification) especially for those cases with multiple types of HPV infection in table 1 could be informative.

7. In table 4, could the authors clarify the data source for the HPV types in each region of Africa if they are from external studies?

8. Could the authors provide a detailed descriptive table as supplement the distribution of HR-HPV types by HIV status for all included women (including the specific types for multiple infections)?

6. PLOS authors have the option to publish the peer review history of their article (what does this mean?). If published, this will include your full peer review and any attached files.

Reviewer #1: No

Reviewer #2: No

---

## [Author Response · Author response to Decision Letter 0]

10 May 2021

Response to reviewer comments: PONE-D-21-04071

28 April 2021

Dear Professor, Joakim Dilner,

Thank you for reviewing our manuscript PONE-D-21-04071- entitled “High-Risk HPV genotypes in Zimbabwean women with cervical cancer: Comparative Analyses between HIV-negative and HIV-positive women” and providing us with constructive criticism on how to improve the statistical analysis and the scientific soundness of the work. Below are our responses to each of the reviewers’ and editor’s comments. 

Reviewer #1:

1. Throughout the text “despite” is often used when it should be “irrespective of” or “regardless of” for example third line in result-section of abstract

Thank you for pointing this out. The authors have edited the document accordingly, and replaced the word “despite” with other terminologies

2. Methods: I would suggest using regression analyses for Table 2, 3 and 4 rather than Chi2. Either Poisson regression for prevalence ratios or Logistic regression for odds ratios.

Thank you for this suggestion. The re-analysis for Tables 2 and 3 was performed, as per the reviewer’s recommendations. Data generated for Table 2 was tested using Poisson’s regression, while Table 3 was analysed using logistic regression. However, Table 4 compares absolute frequencies for the HPV genotypes as recorded by an established database, the HPV Information Centre (https://hpvcentre.net) and so the best analyses to compare these frequencies is using the Chi2 statistical methods.

Table 2: HR-HPV genotypes by HIV status (n=258) and the HPV phylogenetic classification. 

 Frequency (Proportion) 

HPV genotype All HIV negative HIV positive IRR (95% CI) p value

Negative 10 (0.04) 6 (0.02) 4 (0.02) 0.40 (0.22 – 1.00) 0.443

16a 123 (0.48) 66 (0.26) 57 (0.22) 0.97 (0.69 - 1.37) 0.881

18b 65 (0.25) 34 (0.13) 31 (0.12) 1.11 (0.78 – 1.57) 0.573

31a 17 (0.07) 8 (0.03) 9 (0.04) 1.06 (0.57 – 1.96) 0.854

33a 25 (0.10) 18 (0.07) 7 (0.03) 0.81 (0.49 – 1.34) 0.416

35a 68 (0.26) 40 (0.16) 28 (0.10) 0.90 (0.63 – 1.29) 0.574

39b 10 (0.04) 3 (0.01) 7 (0.03) 1.42 (0.67 – 3.05) 0.361

45b 14 (0.05) 8 (0.03) 6 (0.02) 1.05 (0.53 – 2.06) 0.892

51c 11 (0.04) 5 (0.02) 6 (0.02) 1.35 (0.75 – 2.45) 0.316

52a 9 (0.04) 5 (0.02) 4 (0.02) 0.88 (0.33 – 2.38) 0.799

56d 6 (0.02) 2 (0.0) 4 (0.02) 1.11 (0.41 – 2.99) 0.843

58a 23 (0.11) 9 (0.05) 14 (0.05) 1.28 (0.78 – 2.09) 0.334

59b 3 (0.01) 0 (0.00) 3 (0.01) 1.67 (0.53 – 5.24) 0.380

66d 1 (0.0) 0 (0.00) 1 (0.01) 1.1 (0.15 – 7.89) 0.922

68b 7 (0.03) 2 (0.01) 5 (0.02) 1.23 (0.51 – 3.01) 0.645

IRR= incidence rate ratio computed from the Poisson regression analysis 

a= Human papillomavirus alpha 9 species; b= Human papillomavirus alpha 7 species; c= Human papillomavirus alpha 5 species; d= Human papillomavirus alpha 6 species.

Table 3: Multiple HR-HPV infections, stratified by HIV status (n=163).

Number of HPVs found together Frequency (Proportion) 

 All HIV negative HIV positive OR (95% CI) p-value

2 94 (0.59) 45 (0.28) 50 (0.31) 1.65 (0.46 – 5.98) 0.443

3 31 (0.19) 16 (0.10) 15 (0.09) 2.10 (0.55 –8.01) 0.277

4 19 (0.12) 11 (0.07) 8 (0.05) 0.80 (0.16 – 4.02) 0.782

5 12 (0.08) 6 (0.04) 6 (0.04) 1.75 (0.08- 36.28) 0.718

6 6 (0.04) 4 (0.03) 2 (0.01) 1.2 (0.34 – 4.41) 0.756

3. Results: I suggest categorizing by HIV status in Figure 2.

The authors appreciate this recommendation and have presented the data as recommended, please see below.

Figure 2: The HPV mono infection distribution in women with confirmed cervical cancer (n=85).

4. Results: For better understanding of Figure 3A and Figure 3B it might be better if the absolute numbers of certain genotype combinations were presented. I lack an explanation as to how to assess the odds ratios (eg OR increased at 3.3 for the most common combination 16/35 but decreased at OR 0.3 for the second most common combination 16/18. Is 16 reference?). 

Thank you for indicating this, a supplementary table was added to this effect (Supplementary Table 2). Logistic regression analysis was performed to estimate the co-occurrence by adding the two specific HPV genotypes as the predictor and co-variate. For example, where HPV16/18 are analysed, it indicates that HPV16 is the predictor, and HPV18 is the covariate. In the comparative analysis, the reference is the whole dataset, less the HPV genotypes being analysed for co-occurrence. 

Supplementary Table 2: HPV co-occurrence regression analysis, without HIV status, and with HIV status as a co-variate.

Composite Genotypes N (proportion) OR (95% CI)a pa OR (95% CI)b pb

16/18 44 (0.17) 0.3 (0.2- 0.5) <0.001 0.3 (0.2- 0.5) <0.001

16/31 14 (0.05) 0.9 (0.4-2.2) 0.824 1 (0.4- 2.1) 0.773

16/33 18 (0.07) 0.3 (0.2 – 0.6) 0.001 0.3 (0.2-0.7) 0.001

16/35 83 (0.32) 3.3 (1.9 – 5.7) <0.001 3.7 (2.1-6.4) <0.001

16/39 6 (0.02) 0.7 (0.2 – 2.3) 0.555 0.6 (0.2 – 2.9) 0.358

16/45 14 (0.05) 1.7 (0.6 – 4.8) 0.325 1.7 (0.6 – 5.0) 0.331

16/51 15 (0.06) 1.8 (0.6 – 5.2) 0.257 1.6 (0.6 – 4.6) 0.386

16/52 5 (0.02) 0.6 (0.2 – 2.0) 0.391 0.6 (0.2 – 2.2) 0.426

16/56 7 (0.03) 4.2 (0.5 – 34.8) 0.181 4.3 (0.5 – 36.1) 0.182

16/58 23 (0.09) 1.4 (0.6 – 3.2) 0.410 1.4 (0.6 – 3.2) 0.410

16/59 1 (0.00) 0.2 (0.0 – 1.9) 0.154 0.1 (0.0 – 1.3) 0.083

18/31 5 (0.02) 0.6 (0.21 – 1.6) 0.285 0.5 (0.2 – 1.5) 0.250

18/33 12 (0.05) 0.7 (0.4 – 1.5) 0.422 0.8 (0.4 – 1.7) 0.567

18/35 26 (0.10) 0.6 (0.4 – 1.0) 0.068 0.6 (0.4 – 1.1) 0.089

18/39 1 (0.00) 0.2 (0.0 – 1.6) 0.135 0.2 (0.0- 1.3) 0.088

18/45 3 (0.01) 0.4 (0.1 – 1.3) 0.134 0.4 (0.1 – 1.3) 0.116

18/51 7 (0.03) 1.2 (0.4 – 3.0) 0.753 1.0 (0.4 – 2.7) 0.998

18/52 3 (0.01) 0.9 (0.2 – 3.6) 0.899 1.0 (0.2 – 4.0) 0.961

18/56 4 (0.02) 2.2 (0.5 – 8.9) 0.275 2.2 (0.5 – 9.3) 0.291

18/58 11 (0.04) 1.1 (0.5 – 2.5) 0.745 1.0 (0.5 – 2.2) 0.979

18/59 2 (0.01) 2.2 (0.3 – 15.6) 0.443 1.7 (0.2 – 12.5) 0.621

18/68 1 (0.00) 0.3 (0.0 – 2.1) 0.207 0.2 (0.0 – 1.8) 0.164

31/33 1 (0.00) 0.3 (0.0 – 2.1) 0.210 0.3 (0.0 – 1.9) 0.183

31/35 7 (0.03) 0.9 (0.3 – 2.2) 0.751 0.9 (0.4 – 2.3) 0.823

31/45 3 (0.01) 0.3 (0.0 - 1.9 ) 0.183 2.5 (0.7 -9.7) 0.169

31/51 2 (0.01) 1.4 (0.3 – 6.6) 0.647 1.2 (0.3 – 5.8) 0.784

31/52 3 (0.01) 6.0 (1.4 – 25.1) 0.014 6.8 (1.6 – 29.2) 0.010

31/56 1 (0.00) 1.8 (0.2 – 15.5) 0.582 1.8 (0.2 – 15.4) 0.608

31/58 1 (0.00) 0.4 (0.0 – 2.9) 0.348 0.3 (0.0 – 2.6) 0.291

31/68 1 (0.00) 1.6 (0.2 – 13.3) 0.668 1.5 (0.2 – 12.4) 0.734

33/35 8 (0.03) 0.4 (1.2 - 0.9) 0.019 0.4 (0.2 -0.9) 0.024

33/39 1(0.00) 0.6 (0.1 – 4.7) 0.610 0.5 (0.1 – 3.9) 0.494

33/45 4 (0.02) 1.6 (0.5 – 5.2) 0.404 1.6 (0.5 – 5.3) 0.418

33/51 2 (0.01) 0.6 (0.1 – 2.9) 0.560 0.5 (0.1 – 2.5) 0.432

33/52 1 (0.00) 0.6 (0.1 – 5.3) 0.685 0.7 (0.0 – 5.6) 0.720

33/58 5 (0.02) 1.1 (0.4 – 3.0) 0.845 1.0 (0.4 – 2.7) 0.977

33/68 3 (0.01) 2.9 (0.7 – 12.6) 0.150 3.1 (0.7 – 12.9) 0.120

35/39 2 (0.01) 0.4 (0.1 – 2.0) 0.288 0.3 (0.1 -1.7) 0.185

35/45 5 (0.02) 0.4 (0.1 – 2.0) 0.288 0.7 (0.2 -2.0) 0.462

35/51 13 (0.05) 4.1 (1.6 – 10.6) 0.004 4.0 (1.4 – 10.0) 0.008

35/52 2 (0.01) 0.5 (0.1 – 2.3) 0.369 0.5 (0.1 – 2.5) 0.394

35/56 3 (0.01) 1.2 (0.3 – 5.1) 0.805 1.1 (0.3 – 5.1) 0.851

35/58 12 (0.05) 1.2 (0.6 – 2.6) 0.608 1.1 (0.5 – 2.4) 0.832

35/66 1 (0.00) 2 (0.1 – 32.3) 0.625 2.0 (0.1 – 35.0) 0.639

39/45 1 (0.00) 1.6 (0.2 – 12.9) 0.677 1.5 (0.2 -12.9) 0.690

39/51 3 (0.01) 6.2 (1.5-25.7) 0.011 5.5 (1.3 – 23.1) 0.019

39/56 2 (0.01) 11 1.9 -62) 0.007 10.9 (2.0 - 64.3) 0.008

39/58 1 (0.00) 0.9 (0.1 -7.0) 0.894 1 (0.1 – 6.3) 0.810

39/59 1 (0.00) 9.9 (0.9 - 103) 0.056 7.8 (0.7 – 84.6) 0.090

39/68 1 (0.00) 3.4 (0.4 – 30.1) 0.279 3.6 (0.5 – 30.0) 0.280

45/52 1 (0.00) 1.9 (0.2 – 15.9) 0.569 1.9 (0.2 – 15.9) 0.569

45/56 1 (0.00) 2.2 (0.2 – 19.3) 0.479 2.2 (0.2 – 19.3) 0.479

45/58 1 (0.00) 0.4 (0.0 – 3.2) 0.400 0.4 (0.0 – 3.2) 0.400

45/59 1 (0.00) 4.2 (0.4 – 43.8) 0.229 4.2 (0.4 – 43.8) 0.229

45/68 1 (0.00) 2.0 (0.2 - 16.6) 0.533 1.8 (0.2 – 15.6) 0.591

51/52 1 (0.00) 1.6 (0.2 – 13.7) 0.643 1.4 (0.2 – 12.0) 0.740

51/56 2 (0.01) 5.3 (1.0 – 28.0) 0.051 5.3 (1.0 – 28.9) 0.056

51/58 2 (0.01) 1.0 (0.2 – 4.4) 0.965 0.9 (0.2 – 3.9) 0.847

58/68 2 (0.01) 2.3 (1.0 – 11.2) 0.300 2.4 (0.5- 12.5) 0.296

59/68 2 (0.01) 42 (5.3 – 349.4) 0.001 35.4 (4.2 - 301) 0.001

66/68 1 (0.00) 37 (2.2 – 565.6) 0.013 40 (2-766.4) 0.014

a- analyses with no consideration of HIV status; b- analyses with HIV status as a co-variate

5. Results: It says 58% having 2 HR-HPVs in the text but 59% in Table 3.

Thank you for this observation, this statement has been clarified based on re-analysis of the data, and the actual value is 59%.

6. Supplementary Table 1. Suggest adding HIV as risk factor

The authors appreciate this recommendation and have presented the data as recommended.

Supplementary Table 1: Risk factors of HPV in association with the various HPV genotypes

 Odds ratio 95% CI P value

HPV16 

Age 1.0 0.9 – 1.0 0.686

Sexual debut 0.8 0.6 - 1.2 0.304

Parity 1.7 0.8 – 3.6 0.146

STI history 0.7 0.1 – 7.7 0.751

HIV 1.0 0.6 – 1.5 0.839

HPV18 

Age 12.5 -13 – 38 0.333

Sexual debut -6.7 -22 – 9 0.398

Parity 1.3 0.5 – 3.5 0.627

STI history 0.2 -0.0 – 0.09 0.489

HIV 1.2 0.8 – 1.9 0.446

HPV31 

Age 1.0 1.0 – 1.1 0.808

Sexual debut 0.9 0.7 – 1.1 0.340

Parity 0.8 0.6 – 1.0 0.063

STI history 2.3 0.8 – 6.3 0.117

HIV 0.8 0.3 – 2.3 0.664

HPV33 

Age 1.0 0.9 – 1.0 0.391

Sexual debut 1.0 0.8 – 1.2 0.746

Parity 1.0 0.8 – 1.3 0.668

STI history 1.2 0.5 – 3.0 0.684

HIV 0.4 0.2 – 1.1 0.079

HPV35 

Age 1.2 0.9 – 1.5 0.341

Sexual debut 0.7 0.3 – 1.4 0.312

Parity 1.0 -4.0 – 6.0 0.683

STI history 0.0 -0.0 – 0.10 0.576

HIV 0.8 0.5 – 1.3 0.447

HPV39 

Age 1.0 1.0 – 1.1 0.307

Sexual debut 1.2 0.9 – 1.4 0.148

Parity 1.0 0.7 – 1.3 0.761

STI history 1.4 0.4 – 5.1 0.649

HIV 2.1 0.5 – 8.8 0.295

HPV45 

Age 1.1 0.82 – 1.44 0.551

Sexual debut 0.9 0.60 – 1.24 0.414

Parity 1.2 0.45 – 3.39 0.681

STI history 0.3 -0.76 – 1.33 0.566

HIV 1.1 0.4 – 2.8 0.854

HPV51 

Age 1.1 0.88 -1.31 0.480

Sexual debut 0.7 0.25 – 1.7 0.374

Parity 1.1 0.36 – 3.54 0.834

STI history 1.0 -0.24 – 0.52 0.424

HIV 1.9 0.7 - 4.7 0.179

HPV52 

Age 1.1 0.85 – 1.52 0.382

Sexual debut 0.9 0.50 – 1.50 0.579

Parity 0.5 0.09 – 3.18 0.487

STI history 0.3 -0.21 – 0.71 0.242

HIV 0.8 0.2 – 2.9 0.731

HPV58 

Age 1.0 1.0 – 1.1 0.209

Sexual debut 1.0 0.9 – 1.2 0.800

Parity 1.0 0.8 – 1.2 0.661

STI history 0.8 0.3 – 2.0 0.573

HIV 2.5 1.0 – 6.2 0.051

HPV58 

Age 1.1 0.86 – 1.44 0.419

Sexual debut 0.7 0.25 – 1.67 0.374

Parity 1.1 0.44 – 2.79 0.822

STI history 0.0 -0.14 – 0.23 0.611

HIV 1.6 0.8 – 3.4 0.194

HPV59 

Age 0.1 -0.08 – 0.24 0.198

Sexual debut 1.0 0.88 – 1.22 0.650

Parity -0.8 -2.53 – 1.03 0.272

STI history 0.7 -0.70 – 2.03 0.219

HIV 3.7 0.4 – 35.72 0.262

HPV66a 

Age 1.1 0.81 – 1.57 0.473

Sexual debut 1.0 0.81 – 1.21 0.959

Parity 1.0 -0.31 – 0.54 0.369

HIV 1.2 0.1 – 19.47 0.894

HPV68 

Age 1.1 0.83 – 1.39 0.594

Sexual debut 1.1 0.91 – 1.29 0.382

Parity 1.2 0.49 – 2.94 0.685

STI history 0.3 -0.61 – 1.28 0.420

HIV 1.5 0.40 – 5.79 0.535

Number of HPV infections 

Age 1.0 0.95 – 1.04 0.802

Sexual debut 0.9 0.82 – 1.01 0.085

Parity 1.0 0.76 – 1.20 0.690

STI history 0.5 0.14 – 1.62 0.239

HIV 1.5 0.42 – 5.13 0.545

Note: STI, sexually transmitted infection; CI, confidence interval; a – no individual with HPV genotype had STI history

Reviewer #2:

1. It would be informative to provide a brief description of the HPV vaccination program and uptake in Zimbabwe early in the introduction.

The coverage for the HPV prophylactic vaccines is in its infancy in most developing countries, where the coverage is estimated to be more than 12-fold lower than developed countries [10]. As a matter of interest countries such as Zambia, Malawi and Zimbabwe, where the highest global cervical cancer cases are recorded, are still in the pilot and early implementation phases of the national HPV vaccination programs [11-12]. 

2. Could the authors clarify the reasons of only including cervical cancer diagnosed at stage 1b-4.

Thank you for noting this, we have clarified the selection criteria for our cohort. Our study recruited individuals from an outpatient radiotherapy and chemotherapy treatment facility, therefore, only patients seeking this type of care attend the clinics. Consequently, this excludes patients receiving surgical interventions only, or are receiving palliative care, because these individuals receive care from other oncological units in the hospital. The methods describe this as:

The participants from the current study were nested in a pharmacogenomics of cervical cancer study, aiming to recruit women who were newly diagnosed with cervical cancer, and were eligible to receive radical therapy. Potential study participants were identified through the RTC hospital registry. To be considered eligible for this study, women had to be ≥18 years, with histologically confirmed diagnoses of invasive cervical cancer staged between 1b – 4A, and were earmarked for curative anti-cancer therapy. Individuals who were diagnosed with resectable (stage 1A) disease that did not require chemo/radio- therapies, or with advanced disease that required palliative care (stage 4B) were excluded from the study, because these individuals were referred for care outside of the RTC.

3. It seems the histopathological characteristics are available for all cases included in this study. Such information should be included in the descriptive analysis

This descriptive data is now included in the demographics table (Table 1) as illustrated below. 

Tumour Histology Combined n(prop) HIV – n(prop) HIV + n(prop) P

Squamous Cell 207 (0.80) 115 (0.83) 92 (0.77) Ref.

Adenocarcinoma 25 (0.10) 14 (0.10) 11 (0.09) 0.950b

Adenosquamous 11 (0.04) 2 (0.01) 9 (0.08) 0.017 b

Other* 15 (0.06) 8 (0.06) 7 (0.06) 0.900b

Other*= spindle cell carcinoma, papillary serous carcinoma, adenoid cystic, adenosarcoma, small cell carcinoma.

4. Maybe only relevant covariables to the current study are needed to be described. Some of the information collected through the project but not used for the study might not be necessary to mention, such as weight, height, comorbidities and treatment etc.

Thank you for this observation. All variables that are not further analysed in this study have been deleted. The amended statement reads: 

Demographic information such as age, residency, histopathological tumour characteristics, were collected from the patient folder in the RTC. In addition, behavioral and lifestyle factors were collected by interviewing the study participants, namely history of alcohol, smoking and parity. Sexual history information including age at sexual debut, number of sexual partners, history of circumcised partner, sexually transmitted infections were also collected. Data on HIV status was also collected for comparative analysis.

5. I figure no significant association was detected in the univariate regression between HPV-related risk factors (i.e., age, sexual debut, parity and STI history) and any HR-HPV genotypes might be due to limited power, when the HR-HPV types were stratified by each type. However, those are known factors that associated with HPV infection. Could the authors provide estimates (figure 3b) additionally adjusting for those factors and histology at least as sensitivity analysis when examine the co-occurrence of HPV types. Besides, please clarify what is the reference group for the ORs.

As observed by the reviewer, these analyses had not been performed due to a lack of association detected in the univariate analysis. To ensure that these findings were not influenced by limited power of study, multivariable regression was performed using the HPV co-occurrences as predictors, and adjusting for the known HPV risk factors- age, sexual debut, parity, STI history and HIV status, and the findings were tabulated into Supplementary Table 3. Although there were no definitive indicators determined in this sensitivity analyses, we did observe a tendency towards of the HPV16/18 co-occurrence to be influences by age HIV status and age of sexual debut. A larger cohort would be needed to confirm the validity of the potential trend observed here.

Supplementary Table 3. Multivariate regression analysis of co-occurring HPV genotypes with risk factors.

Co-variates OR (95% CI) P

HPV16/18 

Age 1.0 (1.0 – 1.1) 0.684

HIV 2.3 (0.9 -5.6) 0.065

STI history 0.5 (0.2 – 1.3) 0.159

Parity 1.1 (0.9 – 1.4) 0.274

Age of sexual debut 1.1 (1.0 – 1.3) 0.076

HPV16/33 

Age 1.0 (0.9 – 1.0) 0.547

HIV 0.4 (0.1 – 1.5) 0.174

STI history 1.0 (0.2 – 3.9) 0.956

Parity 1.0 (0.7 – 1.3) 0.821

Age of sexual debut 1.0 (0.8 – 1.2) 0.834

HPV16/35 

Age 1.0 (1.0 – 1.1) 0.152

HIV 0.8 (0.4 – 1.6) 0.573

STI history 1.1 (0.7 – 2.2) 0.769

Parity 1.0 (0.9 – 1.2) 0.985

Age of sexual debut 1.0 (0.9 – 1.2) 0.532

HPV35/51 

Age 1.0 (1.0 – 1.1) 0.282

HIV 1.0 (0.2 – 4.4) 0.999

STI history 3.0 (0.8 – 12.2) 0.115

Parity 0.9 (0.7 – 1.3) 0.670

Age of sexual debut 1.0 (0.8 – 1.3) 0.856

HPV35/52 

Age 1.0 (0.8 – 1.3) 0.930

HIV NA NA

STI history NA NA

Parity 0.9 (0.3 – 3.4) 0.933

Age of sexual debut 1.5 (0.7 – 3.6) 0.332

HPV39/51 

Age 1.0 (0.8 – 1.2) 0.827

HIV NA NA

STI history NA NA

Parity 1.2 (0.7 – 2.0) 0.602

Age of sexual debut 1.0 (0.5 – 1.9) 0.984

HPV39/56 

Age 1.2 (1.0 – 1.4) 0.108

HIV NA NA

STI history NA NA

Parity 1.2 (0.7 – 2.3) 0.520

Age of sexual debut 1.2 (0.7 – 2.3) 0.582

6. A legend explaining how the HR-HPV types were classified (ie. hierarchical classification) especially for those cases with multiple types of HPV infection in table 1 could be informative.

Thank you for this recommendation. In our analysis, HPVs were only analysed at a genotype level, due to the limited coverage of the HPV genotyping assay used to test for HPV DNA. We have clarified this in the methods section of the manuscript, and have also included a figure legend in the results for reference to phylogenetic clades in the discussion section.

Methods section:

For analysis all individual HPV genotypes regardless of whether they occur as single or multiple infections were analysed were reported as standalone genotypes. None of the statistical analysis taking into account the species from which the HPV genotypes are from.

Results section:

a= Human papillomavirus alpha 9 species; b= Human papillomavirus alpha 7 species; c= Human papillomavirus alpha 5 species; d= Human papillomavirus alpha 6 species.

7. In table 4, could the authors clarify the data source for the HPV types in each region of Africa if they are from external studies?

Thank you for this comment. We have clarified in the results section where we obtained the data from. The prevalence of the various HPV genotypes in the Zimbabwean study cohort was compared to the most common HPV genotypes on other African women across the continent, using data obtained from HPV Information Centre database (https://hpvcentre.net).

8. Could the authors provide a detailed descriptive table as supplement the distribution of HR-HPV types by HIV status for all included women (including the specific types for multiple infections)?

The additional data for the HPV multiple infections by HIV status was collated into a supplementary table 2. Further inclusion of more than 2 HR-HPVs

---

## [Decision Letter · Decision Letter 1]

12 Aug 2021

PONE-D-21-04071R1

High-Risk HPV genotypes in Zimbabwean women with cervical cancer: Comparative Analyses between HIV-negative and HIV-positive women.

PLOS ONE

Dear Dr. Dandara,

Thank you for submitting your manuscript to PLOS ONE. After careful consideration, we feel that it has merit but does not fully meet PLOS ONE’s publication criteria as it currently stands. Therefore, we invite you to submit a revised version of the manuscript that addresses the points raised during the review process.

Please take into account the concerns raised by Reviewer #2 after the first round of revision.

We look forward to receiving your revised manuscript.

Kind regards,

Graciela Andrei

Academic Editor

PLOS ONE

Journal Requirements:

Reviewers' comments:

Reviewer's Responses to Questions

**Comments to the Author**

1. If the authors have adequately addressed your comments raised in a previous round of review and you feel that this manuscript is now acceptable for publication, you may indicate that here to bypass the “Comments to the Author” section, enter your conflict of interest statement in the “Confidential to Editor” section, and submit your "Accept" recommendation.

Reviewer #1: All comments have been addressed

Reviewer #2: (No Response)

2. Is the manuscript technically sound, and do the data support the conclusions?

Reviewer #1: (No Response)

Reviewer #2: Yes

3. Has the statistical analysis been performed appropriately and rigorously? 

Reviewer #1: (No Response)

Reviewer #2: Yes

4. Have the authors made all data underlying the findings in their manuscript fully available?

Reviewer #1: (No Response)

Reviewer #2: No

5. Is the manuscript presented in an intelligible fashion and written in standard English?

Reviewer #1: (No Response)

Reviewer #2: Yes

6. Review Comments to the Author

Reviewer #1: (No Response)

Reviewer #2: Thank you for the extensive work from the authors in addressing the comments, and manuscript has been substantially improved. However, there are a couple responses to the comments could be further clarified:

In the response to comment point 5, the author has already performed a multivariate regression analysis, using the HPV genotypes as predictors, while controlling for age and HIV status, and the ORs for significant HPV genotypes were illustrated in fig 3b. Therefore, the sensitivity analysis is expecting to be repeating exact the same regression model by additionally control for factors of sexual debut, parity, STI history and histology and presented the ORs as a supplementary figure or table instead of presenting ORs for each of the covariate as shown in the current supplementary table 3.

I appreciate the work done by the authors for supplementary table 2, however, what would be informative to address the comments point 8 is to provide a simple frequency table showing the distribution of hrHPV types, including both single type infection and multiple types of infection by HIV status (HIV positive and HIV negative). Considering there are a proportion of women had more than 2 types of HPV infection (table 2), showing the actual HPV types by HIV status could be interesting to the readership.

7. PLOS authors have the option to publish the peer review history of their article (what does this mean?). If published, this will include your full peer review and any attached files.

Reviewer #1: No

Reviewer #2: No

---

## [Author Response · Author response to Decision Letter 1]

24 Aug 2021

Response to Reviewer Comments: PONE-D-21-04071R1

24 August 2021,

Dear Professor Graciela Andrei

Thank you for reviewing our manuscript entitled “High-Risk HPV genotypes in Zimbabwean women with cervical cancer: Comparative Analyses between HIV-negative and HIV-positive women” and providing us with constructive criticism on how to improve the statistical analysis and the scientific soundness of the work. Below are our responses to each of the reviewers’ and editor’s comments; 

Reviewer #2: Thank you for the extensive work from the authors in addressing the comments, and manuscript has been substantially improved. However, there are a couple responses to the comments could be further clarified. In the response to comment point 5, the author has already performed a multivariate regression analysis, using the HPV genotypes as predictors, while controlling for age and HIV status, and the ORs for significant HPV genotypes were illustrated in fig 3b. Therefore, the sensitivity analysis is expecting to be repeating exact the same regression model by additionally control for factors of sexual debut, parity, STI history and histology and presented the ORs as a supplementary figure or table instead of presenting ORs for each of the covariate as shown in the current supplementary table 3.

Thank you for further clarifying the expectations of the reviewer, which the authors may have misunderstood in the previous peer review process. The authors have done the best they can to address this comment. To this effect, S3 Table was added, consisting of multivariate analysis for the co-segregation of the HPV genotypes, taking into consideration the proposed covariates, namely, age, history of STI, parity, HIV status, age of sexual debut, tumour histology, as suggested. The comment was addressed in the manuscript to read: 

In order to ensure that lack of significance observed was not a result of low study power, a sensitivity appraisal was conducted, using multivariate regression analyses controlling for age, history of sexually transmitted infections, parity, HIV status, age at sexual debut and tumour histology. In this sensitivity analyses, all the co-segregation patterns remained statistically significant except HPV16/33 (OR=0.4; 95% CI=0.2-1.9; p=0.05), HPV35/52 (OR=0.5; 95% CI=0.1-4.3; p=0.50) (S3 Table). 

Table S3. Sensitivity analysis for HPV co-segregation.

HPV genotypes* OR (95% CI) P

16/18 0.3 (0.1-0.5) <0.01

16/33 0.4 (0.2-1.0) 0.05

16/35 3.6 (1.8-7.2) <0.01

35/51 8.8 (2.3-34.3) <0.01

35/52 0.5 (0.1-4.3) 0.50

39/51 4.2 (0.7-25.9) 0.12

39/56 9.5 (1.1-82.9) 0.04

*multivariate analysis for the co-segregation of HPV genotypes, taking into consideration age, history of STI, parity, HIV status, age of sexual debut, tumour histology. 

I appreciate the work done by the authors for supplementary table 2, however, what would be informative to address the comments point 8 is to provide a simple frequency table showing the distribution of hrHPV types, including both single type infection and multiple types of infection by HIV status (HIV positive and HIV negative). Considering there are a proportion of women had more than 2 types of HPV infection (table 2), showing the actual HPV types by HIV status could be interesting to the readership.

Thank you for indicating which analyses and data would be more interesting to the readers. The comments were received well and taken into consideration. The authors have amended the table to report on absolute frequencies, by HIV status, as shown below. This tabulation was favoured as it clearly represents the number of individuals with specific HPV genotypes depending on HIV status.

S2 Table. The number of individuals harbouring specific HPV genotypes stratified by HIV status.

 Number of HPVs (n)

HPV genotype HIV status 1

 2 3 4 5 6

16 - 19 18 5 1 0 0

 + 15 13 3 1 1 0

18 - 13 12 1 0 1 1

 + 6 13 1 0 0 0

31 - 1 0 1 2 0 0

 + 1 0 0 0 0 1

33 - 2 0 1 2 0 0

 + 2 0 0 0 0 0

35 - 15 15 3 0 1 0

 + 3 10 3 0 1 1

39 - 0 2 0 1 0 0

 + 0 1 0 0 0 0

45 - 1 0 2 3 0 0

 + 0 0 1 1 0 0

51 - 0 0 1 0 1 0

 + 1 1 1 1 1 1

52 - 1 2 1 0 0 0

 + 0 3 1 0 0 1

56 - 0 0 0 1 0 0

 + 0 0 1 0 1 0

58 - 0 2 1 1 1 0

 + 0 2 0 0 1 0

59 - 0 0 0 2 0 0

 + 2 0 1 1 0 0

66 - 0 0 0 1 0 0

 + 0 0 1 0 0 0

68 - 2 0 1 0 1 0

 + 1 0 1 1 2 1

TOTAL 85 94 31 19 12 6

---

## [Decision Letter · Decision Letter 2]

31 Aug 2021

High-Risk HPV genotypes in Zimbabwean women with cervical cancer: Comparative Analyses between HIV-negative and HIV-positive women.

PONE-D-21-04071R2

Dear Dr. Dandara,

We’re pleased to inform you that your manuscript has been judged scientifically suitable for publication and will be formally accepted for publication once it meets all outstanding technical requirements.

Kind regards,

Graciela Andrei

Academic Editor

PLOS ONE

Additional Editor Comments (optional):

Reviewers' comments:

Reviewer's Responses to Questions

**Comments to the Author**

1. If the authors have adequately addressed your comments raised in a previous round of review and you feel that this manuscript is now acceptable for publication, you may indicate that here to bypass the “Comments to the Author” section, enter your conflict of interest statement in the “Confidential to Editor” section, and submit your "Accept" recommendation.

Reviewer #1: All comments have been addressed

Reviewer #2: All comments have been addressed

2. Is the manuscript technically sound, and do the data support the conclusions?

Reviewer #1: (No Response)

Reviewer #2: (No Response)

3. Has the statistical analysis been performed appropriately and rigorously? 

Reviewer #1: (No Response)

Reviewer #2: (No Response)

4. Have the authors made all data underlying the findings in their manuscript fully available?

Reviewer #1: (No Response)

Reviewer #2: (No Response)

5. Is the manuscript presented in an intelligible fashion and written in standard English?

Reviewer #1: (No Response)

Reviewer #2: (No Response)

6. Review Comments to the Author

Reviewer #1: (No Response)

Reviewer #2: (No Response)

7. PLOS authors have the option to publish the peer review history of their article (what does this mean?). If published, this will include your full peer review and any attached files.

Reviewer #1: No

Reviewer #2: No

---

## [Editor Report · Acceptance letter]

16 Sep 2021

PONE-D-21-04071R2 

High-Risk HPV genotypes in Zimbabwean women with cervical cancer: Comparative Analyses between HIV-negative and HIV-positive women. 

Dear Dr. Dandara:

I'm pleased to inform you that your manuscript has been deemed suitable for publication in PLOS ONE. Congratulations! Your manuscript is now with our production department. 

Kind regards, 

on behalf of

Dr. Graciela Andrei 

Academic Editor

PLOS ONE